# Past, Current and Future of Fish Diversity in the Alakol Lakes (Central Asia: Kazakhstan)

Nadir Mamilov [1,*], Sayat Sharakhmetov [1,2], Fariza Amirbekova [3], Dinara Bekkozhayeva [4], Nazym Sapargaliyeva [1], Gulnar Kegenova [1], Ainur Tanybayeva [1] and Kanatbek Abilkasimov [5]

[1] Department of Biodiversity and Bioresources, Faculty of Biology and Biotechnology, Al-Farabi Kazakh National University, Al-Farabi Av., 71, Almaty 050040, Kazakhstan; sharakhmetov@gmail.com (S.S.); sapargalyeva.nazym@gmail.com (N.S.); gkegenova78@gmail.com (G.K.); aina.tan@bk.ru (A.T.)

[2] Institute of Genetics and Physiology, Almaty 050060, Kazakhstan

[3] LLP Fisheries Research and Production Center (FRPC), Almaty 050016, Kazakhstan; faryz-91@mail.ru

[4] Laboratory of Signal and Image Processing, Institute of Complex Systems, Faculty of Fisheries and Protection of Waters, CENAKVA, University of South Bohemia in Ceske Budejovice, Zámek 136, 373 33 Nové Hrady, Czech Republic; dbekkozhayeva@frov.jcu.cz

[5] Alakol State Nature Reserve, Kabanbay Batyr Street 32, Usharal 040200, Kazakhstan; alakol_gpz@mail.ru

\* Correspondence: mamilov@gmail.com

**Abstract:** The aboriginal ichthyofauna of the Balkhash basin consists mainly of endemic fish species. By the end of the last century, indigenous fish species were driven out of Lake Balkhash and the Alakol Lakes remain the largest refuges of aboriginal fish fauna. Knowledge of regularities of the modern distribution of the indigenous fishes is crucial for biodiversity conservation as well as restoring aquatic ecosystems. The modern diversity of fish species was investigated there in this study. Significant changes for the indigenous and some alien fish distributions were revealed in contrast with earlier known data. Canonical correspondence analysis (CCA) was used to study the relationships between habitat characteristics and species abundance. Water mineralization and maximal observed water temperatures were estimated as the main environmental variables in fish distribution at the local scale. Habitat change leads to fish fauna homogenization as a result of rare species extinction and alien penetration. Growing human population and poor water management make the future of the indigenous fishes unpredictable.

**Keywords:** biodiversity conservation; freshwater fishes; indigenous; alien; endemic; Alakol Lakes

## 1. Introduction

Human well-being depends on healthy natural ecosystems supported by specific features and functions of every included species [1,2]. Freshwater is a vital resource for each living organism. The maintenance of the health and resilience of natural freshwater ecosystems is vital for human prosperity. The growing human population has made conservation of freshwater fishes an acute problem around the world [3–6]. The modern alarming speed at which species disappear has been called the planet's sixth mass extinction [7,8]. An analysis of the extinction rate of freshwater fishes in the USA and Europe revealed that in the last century, this was about 112 times higher than expected under natural conditions [9]. Current efforts are clearly insufficient to halt the rapid extinction of species [10,11].

Central Asia is a world region with a lack of freshwater resources. Despite the great importance of fish for maintaining the sustainable functioning of aquatic ecosystems and as a food resource, international knowledge of the fish diversity of Central Asia remains unsatisfactory [10,12]. The Balkhash basin is one of the largest oases located in the center of Asia. Paleogeographic and paleoecological processes isolated it from other water systems at least 10,000 years ago. For this reason, the native fish fauna consists of several species and is characterized by a high degree of endemism [13,14]. The naked osman *Gymnodiptychus dybowskii* (Kessler, 1874), Eurasian minnow *Phoxinus phoxinus* (Linnaeus, 1758), spotted

thicklip loach *Triplophysa strauchii* (Kessler, 1874), Tibetan stone loach *Triplophysa stoliczkai* (Steindachner, 1866) and grey loach *Triplophysa dorsalis* (Kessler, 1872) are widespread fish species. Others such as Balkhash marinka *Schizothorax argentatus* Kessler, 1874, Seven River's minnow *Ph. brachyurus* Berg, 1912, Balkhash minnow *Rhynchocypris poljakowii* (Kessler, 1879), plain loach *T. labiata* (Kessler, 1874), Severtzov's stone loach *T. sewerzowi* (Nikolskii, 1938), and the Balkhash perch *Perca schrenkii* Kessler, 1874 are endemic to the basin.

The Alakol Lakes System is located to the east of the Balkhash Basin. Approximately 600 years ago, as a result of climate change, this system of lakes separated from Lake Balkhash, and since then the natural exchange of fish species between Lake Balkhash and Alakol lakes has been impossible [13]. Alakol Lakes system consists of four lakes with rivers flowing into them and a large area of surrounded wetlands (Table 1). As a result, it is an important place for waterbird conservation [15].

**Table 1.** General characteristics of Alakol lakes [16,17].

| Lake | Surface Area, km$^2$ | Maximum Depth, m | Mineralization, g L$^{-1}$ | Inflows (Average % of Total Coming Water) |
|---|---|---|---|---|
| Sassykkol | 736 | 4.7 | 0.27–2.16 | Tentek (95–100%), Karakol, Aiy |
| Koshkarkol | 120 | 5.8 | 0.85–1.28 | Zheniskesu (wet years only) |
| Alakol | 2650 | 54.0 | 1.20–11.60 | Urzhar (50%), Emel (27%) Katynsu (9%), Yrgayity and Zhamanty (9%), Zhamanotkel (5%) and 10 other |
| Zhalanashkol | 38 | 3.3 | 1.20–5.00 | Has not any |

Commercial fishing has used Balkhash marinka, Balkhash perch and spotted thicklip loach. The maximum catch of the Balkhash marinka in the lakes of the Alakol system exceeded 300 tons per year, and of the Balkhash perch—more than 1000 tons per year [17,18]. Despite the fact that Balkhash perch and marinka were highly valued for their taste, some top managers of the fish industry assessed the fish products of the Balkhash and Alakol basins in the middle of the last century as unsatisfactory. As a result, it was decided to replace the native fish species with commercially more valuable ones. In the 1950s–1980s, more than 26 fish species from the basins of the Caspian, Aral, Kamchatka, China and North America were introduced into the basin [19,20]. As a result, aboriginal fish species disappeared from Lake Balkhash and all its large tributaries [20–23]). Many alien species of fish were also introduced into the basin of the Alakol lakes; however, until the beginning of the 21st century, the aboriginal ichthyofauna was preserved [17,24]. Poor water and fishery management are great problems for the region.

The modern states of population of indigenous Balkhash marinka *Schizothorax argentatus*, Seven river's minnow *Phoxinus brachyurus*, plain loach *Triplophysa labiata* and Sewertzov's loach *T. sewerzowii* were listed as vulnerable (VU) in the Red List of threatened species of International Union for Conservation of Nature (IUCN) [25–28].

New findings indicate that the number of threats and risk of extinction has no straight correlation and requires more investment in identifying how threats and different ecosystem stressors operate together at local scales [29].

The purpose of the investigation was to study changes in the fish diversity in the past, assess the current state and forecast further changes. According to that, we investigated patterns of fish species distribution in different parts of the basin.

## 2. Materials and Methods

### 2.1. Study Area and Sampling

The investigation began in 2014 in the Tentek and Shinzhily rivers, anwas extended to various sites in the basin in 2015–2017 and 2020–2021, in July (Figure 1). According to [30], the ichthyofauna in Lake Sasykkol is represented exclusively by alien species, thus we

excluded it from our studies. All types of electrofishing were prohibited in the Republic of Kazakhstan. Therefore, we used a kick net and a landing net for catching fish. This method is not the best, but it allows providing data on the total diversity, dominant species, and relative abundance of fish [31]. The method is traditionally used in Kazakhstan, and makes it possible to compare data obtained for different water bodies and by different researchers [32,33]. Fish sampling was provided according to special permissions of the local fishery authority. Fishing was carried out over an area of about 100 m$^2$ for each location. The rivers were sampled in such a way as to capture the riffle and pool and the total microhabitats system. In the summer, many rivers of the Alakol basin do not reach the delta. Therefore, the main attention was paid to the transfer zone (middle sections, medium gradient) of the rivers since the greatest diversity of ichthyofauna is concentrated there. The coordinates and short designations of the sampling sites are presented in Table 2. A total of 50 sampling events occurred from 20 localities.

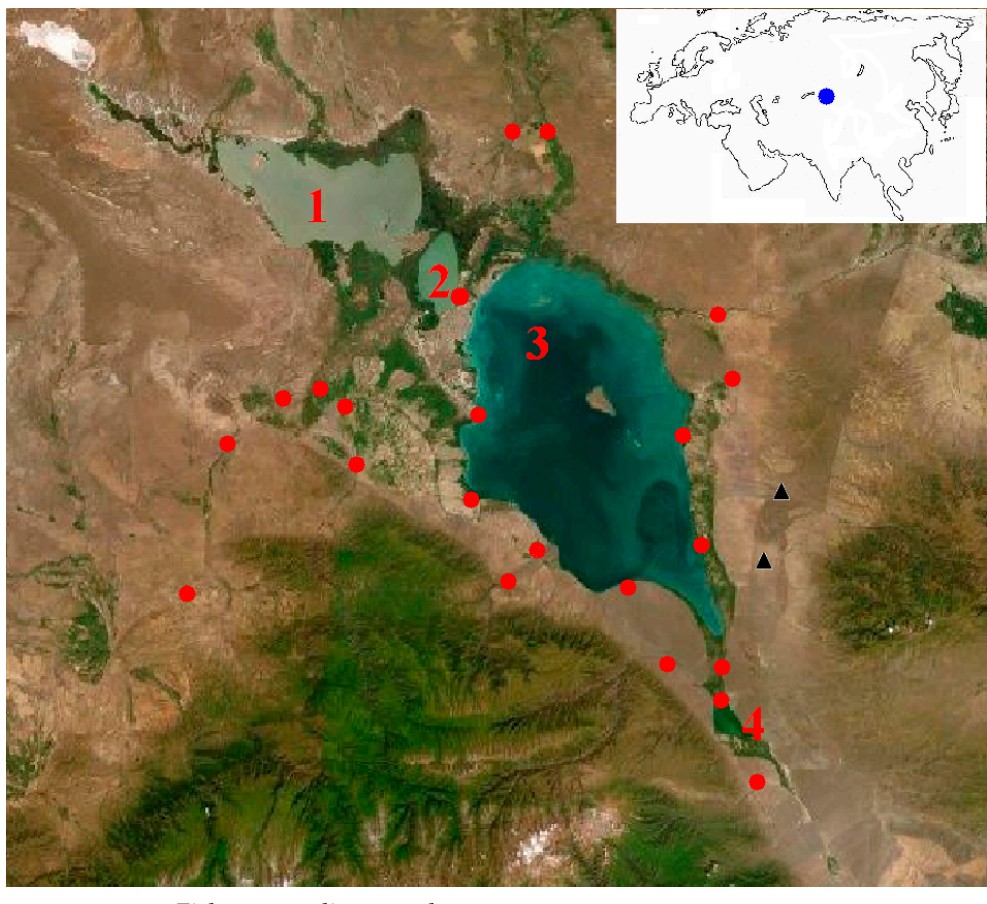

●      Fishes were discovered
▲      Fishes were not discovered

**Figure 1.** Schematic map of the investigated region: 1—Sasykkol Lake, 2—Koshkarkol Lake, 3—Alakol Lake, 4—Zhalanashkol Lake. The blue point shows the position of the Alakol Lakes in Eurasia.

Several abiotic parameters of water were studied at each site: color, odor, turbidity, temperature, pH, ammonium and nitrate content. The color and smell were determined using the senses (organoleptic). All mentioned water characteristics were investigated at the sampling places. Water turbidity was determined using a turbo-performance HI 93703 "Hanna Instruments", salinity, temperature and pH using joint device of the same manufacturer HI 98129, ammonia within HI 96700 ammonia LR, and nitrate with HI 93728.

**Table 2.** Coordinates and abbreviations of sampling sites.

| Site | Coordinates | | Years | Biotopes | Abbreviation |
|---|---|---|---|---|---|
| | N | EO | | | |
| Alakol Lake, near Akshi settlement (recreation area) | 45°54′53″ | 81°37′00″ | 2021 | Lake | Al21a |
| | | | 2021 | Estuary | Al21z |
| | | | 2020 | Lake | Al20a |
| | | | 2020 | Estuary | Al20z |
| | | | 2017 | Lake | Al17a |
| | | | 2016 | Lake | Al16a |
| | | | 2015 | Lake | Al15a |
| Alakol Lake, eastern side | 45°54′53″ | 82°03′37″ | 2020 | Wetland | Al20e |
| | | | 2020 | Lake | Al20l |
| Alakol Lake, near Kabanbay settlement (recreation area) | 46°05′41″ | 82°01′29″ | 2021 | Lake | Al21b |
| | | | 2021 | Wetland | Al21lim |
| Alakol Lake, near Koktuma settlement (recreation area) | 45°50′13″ | 81°55′20″ | 2021 | Wetland | Al21k |
| Alakol Lake, near estuary of Zhamanty River | 45°54′47″ | 81°37′30″ | 2021 | Estuary | Zm21 |
| | | | 2020 | Estuary | Zm20 |
| | | | 2017 | Estuary | Zm17 |
| | | | 2016 | Estuary | Zm16 |
| Sassykkol Lake | 46°40′48″ | 80°35′11″ | 2021 | Wetland | SK21 |
| | | | 2020 | Wetland | SK20 |
| Zhamanty River, headwater zone | 45°51′34″ | 81°24′58″ | 2015 | Straight channel, stream | Zm15t |
| | | | 2016 | | (no fish) |
| | | | 2017, 2020, 2021 | Dry channel | (no water) |
| Tentek River, depositional zone | 46°12′25″ | 80°53′09″ | 2021 | Meandering channel | Tk21 |
| | | | 2020 | | Tk20 |
| | | | 2017 | | Tk17 |
| | | | 2016 | | Tk16 |
| Tentek River, headwater zone | 46°03′17″ | 81°01′37″ | 2021 | Winding channel | Tk21t |
| | 46°03′17″ | 81°01′37″ | 2016 | | Tk16t |
| | 46°03′17″ | 81°01′37″ | 2015 | | Tk15t |
| | 46°03′17″ | 81°01′37″ | 2014 | | Tk14t |
| Right side tributary of Tentek River | 46°12′24″ | 80°51′10″ | 2017 | Meandering channel | Ua17 |
| | | | 2016 | | Ua16 |
| | | | 2015 | | Ua15 |
| Tokty River, transfer zone | 45°26′03″ | 82°15′13″ | 2020 | Winding channel | T20 |
| Shynzhyly River, headwater zone | 45°49′28″ | 80°33′51″ | 2021 | Meandering channel | Sh21t |
| | | | 2020 | | Sh20t |

**Table 2.** *Cont.*

| Site | Coordinates | | Years | Biotopes | Abbreviation |
|---|---|---|---|---|---|
| | **N** | **EO** | | | |
| Shynzhyly River, transfer zone | 45°23′38″ | 80°27′54″ | 2021 | Winding channel | Sh21m |
| | | | 2020 | | Sh20m |
| | | | 2017 | | Sh17m |
| | | | 2016 | | Sh16m |
| | | | 2015 | | Sh15m |
| P.Shynzhyly, depositional zone | 46°12′20″ | 80°52′15″ | 2021 | Meandering channel | Sh21l |
| | 46°12′20″ | 80°52′15″ | 2020 | | Sh20l |
| Urzhar River, transfer zone | 47°3′10″ | 81°32′19″ | 2021 | Meandering channel | U21 |
| | 47°3′10″ | 81°32′19″ | 2020 | | U20 |
| Karakol River, transfer zone | 46°57′53″ | 80°44′32″ | 2021 | Meandering channel | Kr21 |
| Yrgayity River | 45°40′10″ | 82°01′48″ | 2015,2016,2020 | Straight channel, stream | Y20 |
| Emel River, transfer zone | 46°22′46″ | 82°15′16″ | 2021 | Meandering channel | E21 |
| | 46°22′46″ | 82°15′16″ | 2020 | | E20 |
| Shagyntogay River, transfer zone | 46°16′43″ | 82°13′14″ | 2020 | Meandering channel | (no fish) |
| | | | 2021 | Dry | (no water) |
| Zhalanashkol Lake | 45°36′17″ | 82°09′33″ | 2021 | Lake | ZK21 |
| | 45°36′17″ | 82°09′33″ | 2020 | | ZK20 |
| Katynsu River, transfer zone | 46°46′39″ | 82°03′17″ | 2021 | Meandering channel | K21 |
| | 46°46′39″ | 82°03′17″ | 2020 | | K20 |

*2.2. Descriptive Statistics*

The correlation between the abiotic parameters of water and fish species, as well as the pairwise presence/absence of fish species, was calculated from the correlation matrix using the Spearman coefficient in R computer environment vs. 3.0 [34]. Spearman's correlation coefficients are considered significant at 0.5 level and 95% confidence level ($p \leq 0.05$). We also compared the samples by fish species composition using the principal component analysis (PCA) [35] in the NTSYS program version 2.1 and canonical correspondence analysis (CCA) [36] in the R.

## 3. Results

The modern diversity of the fish fauna and data of previous years is presented in Table 3. The most widespread and quite numerous are the spotted thicklip loach, naked osman and Balkhash marinka. Seven River's minnow and Eurasian minnow were encountered once. The Eurasian minnow was found only in the Karakol River and the Seven River's minnow was observed in the wetland on the east side of the Alakol Lake. The minnows were the dominant species in terms of numbers for both of these sites. There were also found to be some loaches, combining the features of gray and Tibetan stone loaches, and plain loach and Tibetan stone loach. Of the alien species, pike perch was represented only in some commercial and game catches. In high-water years, common bream in the Tentek River penetrated far upstream and had been noted in areas with a rapid current, typical of naked osman, plain loach and Tibetan stone loach.

**Table 3.** Fish diversity changes over time.

| Latin Name | English Name | Abbreviation for Revealed Species | Until 1960 [37,38] | 1960–2014 [17,18,24] | Original Data 2015–2017, 2020, 2021 | |
|---|---|---|---|---|---|---|
| | | | | | Number of Samples | Number of Fishes |
| *Indigenous species:* | | | | | | |
| *Phoxinus phoxinus* (Linnaeus, 1758) | Eurasian minnow | *Php* | + | + | 1 | 118 |
| *Phoxinus brachiurus Berg, 1912* | Seven River's minnow | *Phb* | 0 | + | 1 | 28 |
| *Rhynchocypris poljakowii* (Kessler, 1879) | Balkhash minnow | *Pol* | 0 | + | 13 | 282 |
| *Schizothorax argentatus Kessler, 1874* | Balkhsh marinka (snowtrout) | *Sha* | + | + | 17 | 344 |
| *Gymnodiptychus dybowskii* (Kessler, 1874) | Naked osman | *Gyd* | + | + | 24 | 806 |
| *Triplophysa strauchii* (Kessler, 1872) | Spotted thicklip loach | *Trs* | + | + | 28 | 489 |
| *Triplophysa stoliczkai* (Steindachner, 1866) | Tibetan stone loach | *Tst* | + | + | 15 | 157 |
| *Triplophysa dorsalis* (Kessler, 1872) | Gray loach | *Tdo* | + | + | 2 | 131 |
| *Triplophysa labiata* (Kessler, 1874) | Plain loach | *Trl* | + | + | 12 | 278 |
| *Triplophysa sewerzowii* (G.Nikolsky, 1938) | Severtsov's loach | *Tse* | + | + | 18 | 90 |
| *Triplophysa dorsalis × Triplophysa stolickai* | (Hybrid) | *Hy1* | 0 | 0 | 3 | 7 |
| *Triplophysa labiata × Triplophysa stolickai* | (Hybrid) | *Hy2* | 0 | 0 | 1 | 3 |
| *Perca schrenkii* Kessler, 1874 | Balkhash perch | *Per* | + | + | 16 | 448 |
| *Alien species:* | | | | | | |
| *Acipenser ruthenus* Linnaeus, 1758 | Sterlet sturgeon | | + | 0 | 0 | 0 |
| *Cyprinus carpio* Linnaeus, 1758 | Carp | *Cyp* | + | + | 3 | 17 |
| *Abramis brama* (Linnaeus, 1758) | Bream | *Abr* | 0 | + | 4 | 48 |
| *Tinca tinca* (Linnaeus, 1758) | Tench | | + | 0 | 0 | 0 |
| *Ctenopharyngodon idella* (Valenciennes, 1844) | Grass carp | | 0 | + | 0 | 0 |
| *Hypophthalmichthysmolitrix* (Valenciennes, 1844) | Silver carp | | 0 | + until 2000 | 0 | 0 |
| *Hypophthalmichthys (Aristichthys) nobilis* (Richardson, 1845) | Bighead carp | | 0 | + until 2000 | 0 | 0 |
| *Pseudorasbora parva* (Temminck et Schlegel, 1846) | Pseudorasbora, or topmouth gudgeon | *Pse* | 0 | + | 16 | 176 |
| *Carassius gibelio* (Bloch, 1872) | Prussian carp | *Cag* | 0 | + | 10 | 46 |

**Table 3.** *Cont.*

| Latin Name | English Name | Abbreviation for Revealed Species | Until 1960 [37,38] | 1960–2014 [17,18,24] | Original Data 2015–2017, 2020, 2021 | |
|---|---|---|---|---|---|---|
| | | | | | Number of Samples | Number of Fishes |
| *Rhodeus ocellatus* (Kner, 1866) | Rosy bitterling | *Rho* | 0 | 0 | 1 | 1 |
| *Rutilus rutilus* (Linnaeus, 1758) | Roach | *Rut* | 0 | + | 3 | 21 |
| *Parabramis pekinensis* (Basilewsky, 1855) | White amur bream | *Pab* | 0 | 0 | 2 | 3 |
| *Hemiculter leusiculus* (Basilewsky, 1855) | Sharpbelly | | 0 | + | 0 | 0 |
| *Abbottina rivularis* (Basilewsky, 1855) | Abbottina or false gudgeon | *Abb* | 0 | + | 2 | 64 |
| *Gobio cynocephalus* Dybowski, 1869 | Siberian gudgeon | *Gcy* | 0 | 0 | 1 | 31 |
| *Lefua costata* (Kessler, 1876) | Eightbarbel loach | | 0 | + | 0 | 0 |
| *Oncorhnchus mykiss* (Walbaum, 1792) | Rainbow trout | | 0 | + | 0 | 0 |
| *Orizias latipes* (Temminck et Schlegel, 1846)/*Orizias sinensis* Chen, Uwa et Chu, 1989 | Japanese rice fish | *Ors* | 0 | + | 5 | 21 |
| * *Sander lucioperca* (Linnaeus, 1758) | Pike-perch | | 0 | + | + | + |
| *Sander volgensis* (Gmelin, 1789) | Volga pikeperch | | 0 | + | 0 | 0 |
| *Micropercops sinctus* (Dabry de Thiersant, 1872) | | *Mci* | 0 | + | 3 | 9 |
| *Rhinogobius cheni* (Nichols, 1931) | | *Rhs* | 0 | + | 2 | 2 |

* Observed in commercial catches only. 0—species was not revealed, +—species was revealed

Figure 2 shows the habitat groupings of various native and alien species. The joint habitation of Balkhash perch and spotted thicklip loach is natural and corresponds to the data of long-term observations [17,24,37,38]. All other groups combine native and alien species.

The relative position of samples from different rivers in the space of the first to third principal components is shown in Figure 3. According to the species composition, the most detached samples of different years were observed in the lower section of the Tentek River and its tributary, as well as samples from the Emel River in 2020 and different estuaries of Lake Alakol.

The greatest loads on the principal component are exerted by the aboriginal Severtsov's loach, a hybrid of the gray and Tibetan loaches, as well as the alien false gudgeon, bream, white amur bream and medaka. Thus, we did not find any patterns in relation to the origin or ecology of the species.

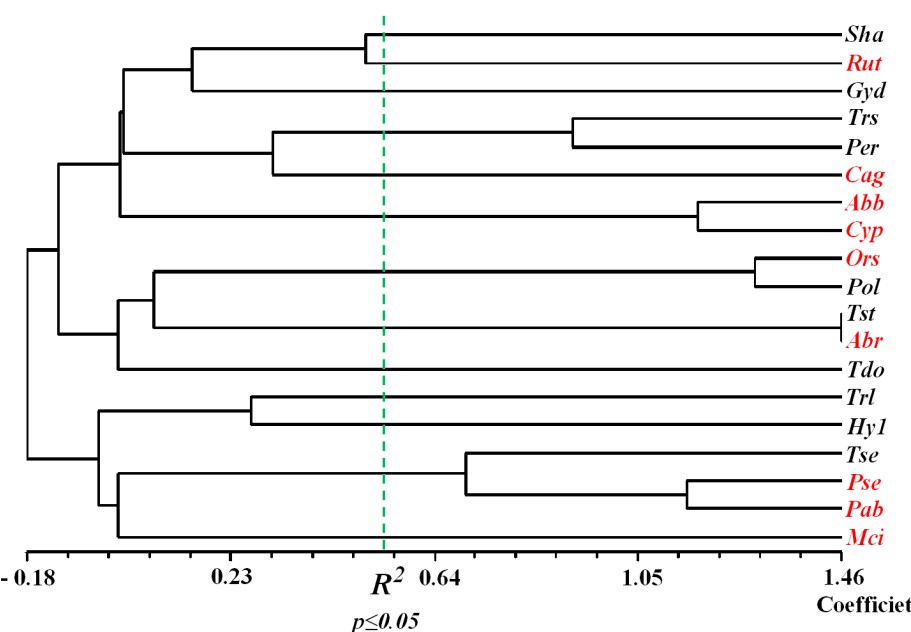

**Figure 2.** Fish interspecies distribution similarity dendrogram (UPGMA, invariants were excluded). The names of the aboriginal species are shown in black; the introduced species are shown in red. The green line shows a significant positive correlation of pairwise comparison (Trs × Per, Abb × Cyp, etc.).

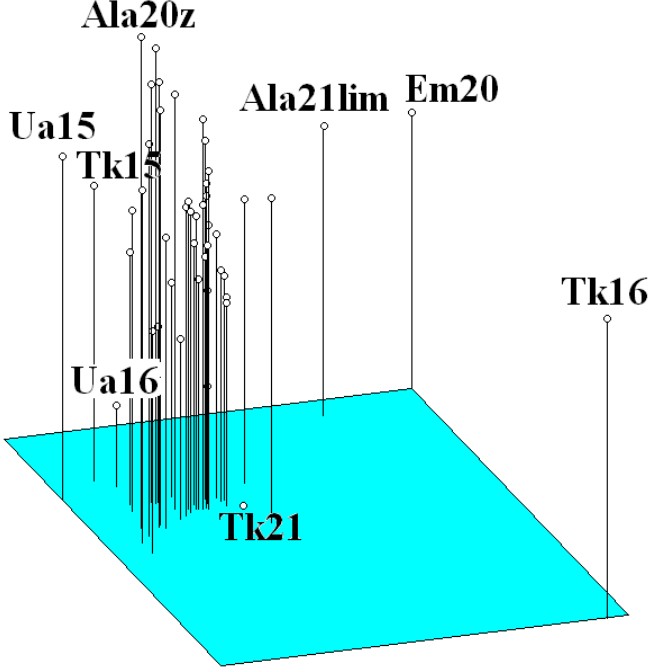

**Figure 3.** Position of samples from different rivers. Designations are given for the most deviating samples: Ala20z—Alakol Lake, estuary, 2020; Ala21lim—Alakol Lake, near Kabanbay settlement, wetland, 2021; Em20—Emel River 2020; Ua15 and Ua16—right side tributary of Tentek River in 2015 and 2016; Tk15, Tk16 and Tk21—Tentek River, depositional zone, 2015, 2016 and 2021.

CCA was used to study possible relationships between five environmental variables (turbidity, temperature, pH, ammonium and nitrate content) and fish abundance in the 47 samples (Table 2). Eigenvalues (lambda) were 0.466, 0.381, 0.197 and 0.116 for the first (horizontal), second (vertical), third and fourth axes, respectively. The first two axes explained 68.1% of the variance and all the canonical axes collectively explained 93.4% of variance. The sites/objects data were linearly related to the sites/variables data by result of permutation test (500 permutations, F = 1.291, P = 0.030). The first axes reflected the distribution of species and sampling sites along with maximal observed water temperature (Figure 4). Along the second axis of variability, sampling sites were ordered by increasing mineralization and concentration of ammonium ions. Results of CCA indicated three patterns in fish distribution according to measured environmental variables (Figure 4). Balkash perch survived in the lake with more salty water. Many alien fish species as topmouth gudgeon, siberian gudgeon, carp, abbottina, white amur bream as well as indigenous Seven River's minnow and Severtsov's loach preferred warm water with low mineralization. Other indigenous and alien species kept to rather cool water.

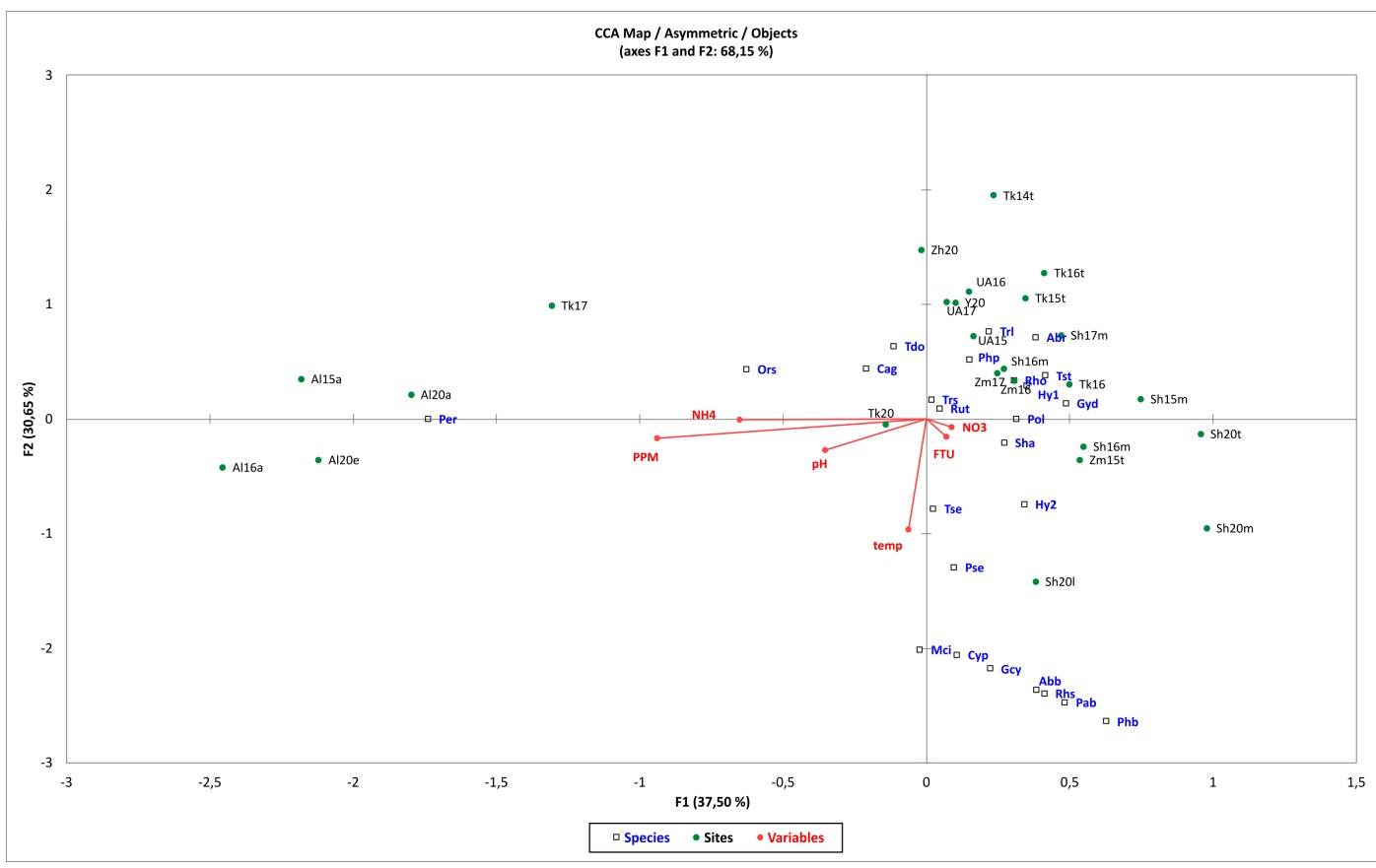

**Figure 4.** CCA plot of species (objects), sites and environmental variables of water bodies in the Alakol Lakes watershed: first axis horizontally, second axis vertically. Abbreviations of sites are given in Table 2 and species in Table 4. Abbreviations of the environmental arrows are PPM—water mineralization, pH—pH value, temp—maximal observed water temperature (°C), FTU—water turbidity (formazin turbidity unit), $NH_4$ and $NO_3$ concentrations of ammonia and nitrates, respectively.

**Table 4.** Principle component factor loadings for the species.

| Fish Species | Abbreviation | Principal Components | | |
|---|---|---|---|---|
| | | **1** | **2** | **3** |
| Indigenous: | | | | |
| *Schizothorax argentatus* | *Sha* | 0.0071 | 0.0843 | 0.2987 |
| *Gymnodiptychus dybowskii* | *Gyd* | −0.0401 | −0.0272 | 0.2695 |
| *Rhynchocypris poljakowii* | *Pol* | −0.0556 | −0.0436 | −0.4802 |
| *Triplophysa labiata* | *Trl* | 0.1469 | 0.1648 | 0.1446 |
| *Triplophysa dorsalis* | *Tdo* | 0.0019 | −0.0068 | −0.1023 |
| *Triplophysa sewerzowii* | *Tse* | 0.3078 | −0.2884 | 0.0728 |
| *Triplophysa stoliczkai* | *Tst* | −0.4990 | −0.3940 | 0.1230 |
| *Triplophysa strauchii* | *Trs* | −0.0317 | 0.0741 | 0.3327 |
| *Triplophysa dorsalis* × *Triplophysa stoliczkai* | *Hy1* | 0.4522 | 0.3757 | −0.1560 |
| *Perca schrenkii* | *Per* | −0.0880 | 0.1030 | 0.1109 |
| *Phoxinus phoxinus* | *Php* | | invariant | |
| *Phoxinus brachiurus* | *Phb* | | invariant | |
| *Triplophysa labiata* × *Triplophysa stoliczkai* | *Hy2* | | invariant | |
| Alien: | | | | |
| *Carassius gibelio* | *Cag* | −0.0522 | 0.0577 | 0.0628 |
| *Cyprinus carpio* | *Cyp* | −0.2241 | 0.2905 | −0.0796 |
| *Pseudorasbora parva* | *Pse* | 0.2222 | −0.3077 | −0.0742 |
| *Abbottina rivularis* | *Abb* | −0.3097 | 0.3986 | −0.0575 |
| *Abramis brama* | *Abr* | −0.3495 | −0.2516 | −0.1354 |
| *Rutilus rutilus* | *Rut* | 0.0086 | 0.0208 | 0.2356 |
| *Parabramis pekinensis* | *Pab* | 0.3097 | −0.3986 | 0.0575 |
| *Orizias sinensis* | *Ors* | −0.0144 | −0.0569 | −0.5579 |
| *Micropercops sinctus* | *Mci* | 0.0120 | −0.0152 | −0.0309 |
| *Gobio cynocephalus* | *Gcy* | | invariant | |
| *Rhodeus ocellatus* | *Rho* | | invariant | |
| *Rhinogobius cheni* | *Rhs* | | invariant | |

In 2020 and 2021, ammonium ions were detected only in different areas of Lake Alakol (Figure 5) and Lake Sasykkol, nitrate ions in various concentrations were found not only in the Lake Alakol (Figure 6) but also in most rivers (presented in attachments). Compared with the data of previous observations [17], the concentration of ammonium is at the level of 2004, and the content of nitrates has increased markedly.

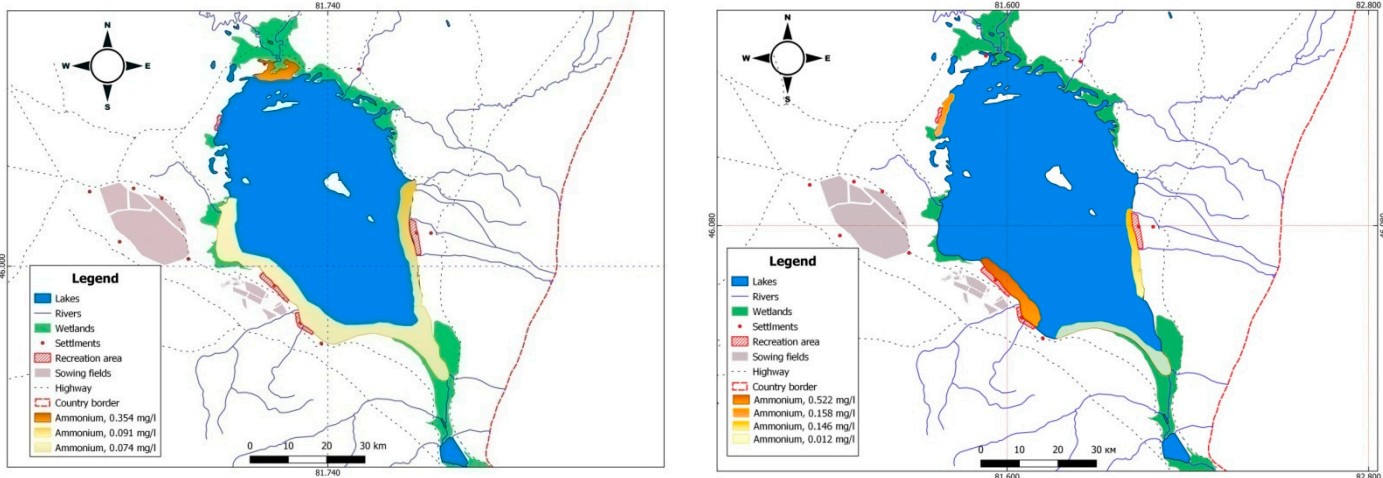

**Figure 5.** Distribution of ammonium ions in Lake Alakol in 2020 (**left**) and 2021 (**right**).

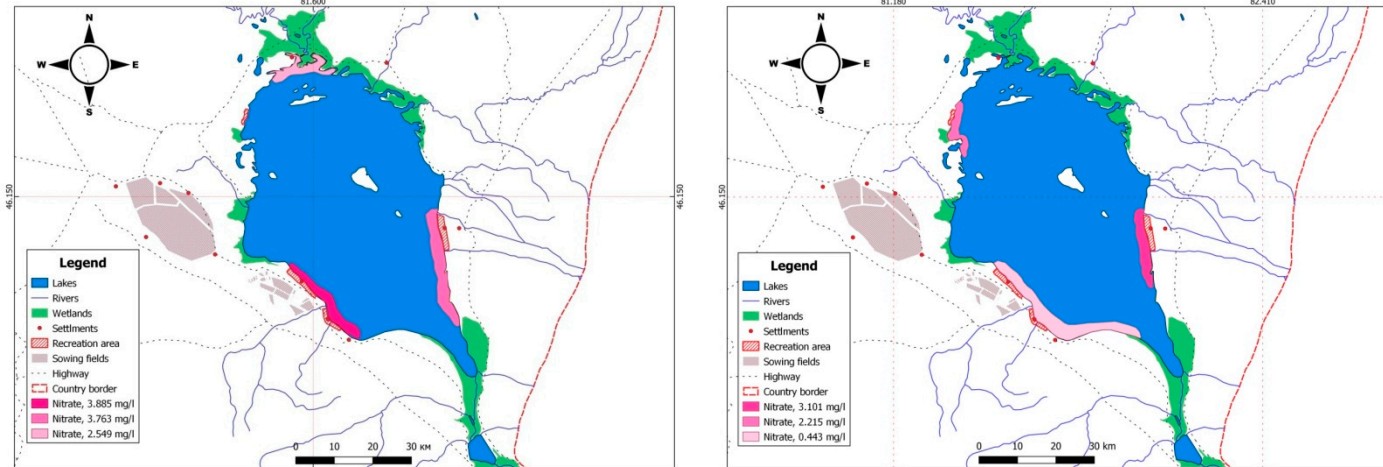

**Figure 6.** Distribution of nitrate ions in Lake Alakol in 2020 (**left**) and 2021 (**right**).

## 4. Discussion

Our results showed that all native species still exist in water bodies of the basin of Alakol lakes, but many of the distribution areas significantly shortened in contrast with previously known data [17,18,24]. Earlier, for the rivers of the Balkhash basin, a natural change in fish diversity from source to mouth was established [13]. The upper zone with a swift current was occupied by naked osman and Tibetan stone loach. Below them were added plain thicklip loach, then spotted thicklip loach, after that Balkhash minnow, Balkhash marinka and Balkhash perch. In the lower zone, the proportion of naked osman, Tibetan stone loach, and plain thicklip loach decreased, and the Seven River's minnow, gray loach, and Severtsov's loach appeared. In the lakes, the main fish species were Balkhash perch, Balkhash marinka and spotted thicklip loach.

At present, the natural flow of all rivers, with the exception of the Yrgayity River, has been disturbed as a result of the water extraction and construction of dams. This type of human impact on the diversity and status of aquatic organisms is extremely complex, and one of the most common ways of interfering with natural freshwater ecological processes around the world. The stream regulation leads to significant abiotic changes in river ecosystems as changes in the temperature, turbidity, speed of stream, and flow/level regimes [3–6,39]. In the basin of the Alakol Lakes, this has also led to a violation of habitat conditions.

Balkhash perch disappeared from all rivers. The number of rivers inhabited by gray loach, plain thicklip loach, Eurasian and Seven River's minnows has decreased.

The Emel River has converted into a hot spot of different alien fish species. The river originates in the territory of the People's Republic of China, where a reservoir has been built and intensive fish-breeding work is underway, including the introduction of new species of fish. Unlike other studied rivers, alien species dominate here in terms of diversity and abundance. The eightbarbel loach *Lefua costata* (Kessler, 1876) was found here in 2015, and then the gudgeon *Gobio cynocephalus* Dybowski, 1869 in 2021. We were surprised to find common bream in cold water as well as Siberian gudgeon in so warm water. The taxonomy of gudgeons inhabiting the water bodies of Kazakhstan needs revision [40]. Therefore, we do not exclude that, upon careful examination, it will turn out to be a different type of gudgeon. Severtsov's loach and Tibetan stone loach from indigenous fish species have survived in the Emel River.

Only indigenous fish species inhabited the middle and lower sections of the Shagynto-gay and Zhamanty rivers about 20 years before [17]. Last year, the sections of both rivers were going to be dry in summertime.

Alien fish species spread in all the lakes and rivers. The problem of replacing native ichthyofauna with alien species is one of the most acute [41,42]. Despite the previously carried out large-scale work on the introduction of new species and transboundary transfer along the Emel River, local fish species remain dominant not only in rivers but also in lakes Alakol and Zhalanashkol. However, their area of distribution is shrinking. Therefore, in not one of the investigated water bodies, have we found the full composition of aboriginal fish species that was described in previous publications [13,17,24,37,38]. Balkhash perch disappeared from rivers, and spotted thicklip loach did not occur in commercial catches in Lake Alakol for the last two years.

We had revealed growth of concentration of nitrogen in the Alakol Lake and presence in different rivers. No statistically reliable correlation between every species distribution and nitrogen concentration was found. In remote water ecosystems, nitrogen comes from agricultural land due to fertilizers, *N*-fixing crops, animal husbandry, sewage and atmospheric deposition due to the combustion of fossil fuels [43,44]. Nitrogen compounds can lead to freshwater acidification and have far-reaching effects on freshwater ecosystems, and so may be an early alarm on rapid and radical changes in the future [43,45].

Habitat change leads to similarity of fish assemblages in terms of species composition called homogenization of fish fauna in freshwater bodies around the world [41]. At this stage, the homogenization of ichthyofauna in the Alakol basin occurs due to the disappearance of local species, which is not typical for most other water bodies. Rare native species, as shown in Table 3, are still avoiding extinction due to their large numbers in suitable habitats [46]. In the basin of the Alakol lakes, the Eurasian minnow was found only in the Karakol River, the Seven River's minnow—in the wetland off the eastern coast of Lake Alakol, gray loach—in the Tentek River and the mouth of the Zhamanty River, plain thicklip loach in the Tentek River and its tributary. Severtsov's loach in some wetlands around Alakol Lake was numerous twice, in 2016 and 2021. The territory of the Alakol nature reserve covers only the mouths of the Tentek and Urzhar rivers and part of the Zhalanshkol lake. This prevents the natural flow of water from being regulated. Previously, fishing was constrained by the poor quality of the roads. In the last decade, recreation areas and agricultural lands began to develop intensively in the basin of the Alakol lakes. Completion of the construction of a high-speed highway is scheduled for 2022. All this will lead to an increase in anthropogenic load.

We have selected a set of measurable and widespread indicators of the negative impact of humans on local fish populations. Of these, only an increase in water temperature in the Emel River clearly favored the existence of many alien species. However, a decrease in living area has been identified, as well as a change in the habitats of many aboriginal species. Generally, our results confirm the new achievement of Greenville et al. [29] that particular combinations of threats are poor predictors of extinction risk on a regional level

and effective conservation require a greater push for investment to identify how threats and different ecosystem stressors operate together at local scales.

The current state of aquatic ecosystems is of great concern due to the growing population in this region, poor water and fishery management, and illegal fishery [47–49]. Therefore, the future of native fish species seems pessimistic.

**Author Contributions:** Conceptualization, methodology, investigations, data analysis, original draft preparation, supervision, N.M.; field investigation and data curation S.S., F.A., D.B., N.S. and G.K.; formal analysis and resources, N.M., A.T. and K.A.; visualization, A.T.; project administration, K.A. All authors have read and agreed to the published version of the manuscript.

**Funding:** Dinara Bekkozhayeva's participation in the project was supported by the Ministry of Education, Youth and Sports of the Czech Republic—project GAJU 013/2019/Z.

**Institutional Review Board Statement:** Ethical review and approval were waived for this study due to the low of the Republic of Kazakhstan "On protection, reproduction and use of the animal world" from 9 July 2004 No593-II.

**Acknowledgments:** We are grateful to Boris Annenkov, Roman and Arthur Skorovs for their logistic support and help for the field investigations, and staff of the Alakol Nature State Reserve for their consultations and participation in the field investigations. We highly appreciate the time, patience, constructive critics and advice of both the anonymous reviewer and academic editor. The authors thank Robert Gentleman, Ross Ihaka and R Development Core team for the free software.

**Conflicts of Interest:** We declare that the fish were caught under a survey permit of local state fishery authority and program of research of Alakol Nature State reserve, and absence of any conflict of interest.

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
