# Peer review of "Past, Current and Future of Fish Diversity in the Alakol Lakes (Central Asia: Kazakhstan)"

_diversity, doi:10.3390/d14010011_

Round 1
Reviewer 1 Report
The manuscript provides information on fish fauna of Alakol Lakes in central Asia and discusses threats to the indigenous species because of introduces species. Authors have also collected environmental parameters for the various sampling sites. Although authors have interesting and valuable data from their studies, they have not performed the analysis adequately. Comments on the manuscript are provided below, based on which authors can modify their analysis and presentation.
(1) Statistical analysis: Authors have environmental data but they use it in a very limited sense by providing cutoff for the R2 value below which the correlations will be significant in the clades identified in UPGMA tree. There are two problems with this analysis. First, if authors are performing pairwise correlations in the analysis then they have to correct for family wise error rate (some correlations will be identified as significant even when they are not). Second, this analysis does not actually explain, how the species distribution and abundance are affected by the different environmental factors. Using Principal component analysis for looking at the similarity and differences among the sites is also interesting analysis (PCA) but does not show how the sites are related based on both species composition as well as environmental parameters. Fortunately, there is just one analysis that authors can perform to replace both the UPGMA tree as well as PCA and provide information on all three aspects, the distribution of environmental parameters and the distribution of species across the various sampling sites. This analysis is called Canonical Correspondence Analysis. Authors will need to first arrange their data with various sampling sites in the rows and environmental parameters (both quantitative and qualitative presence absence can be used) in columns followed by abundance of various species in the sampling sites with a separate column for each species. The data can then be analysed using PAST software (https://www.nhm.uio.no/english/research/infrastructure/past/). Authors can statistically check if there is correlation between the environmental parameters and species abundance using the permutations test provided by the CCA module of the software. If authors plot a triplot of CCA, it will show vectors of environmental parameters and factor loadings for both the species as well as sampling sites with labels.
(2) Data availability: It will be better if authors make their data on environmental parameters and species distribution and abundance in various sampling sites available as supplementary information. This forms part of the reproducibility clause of scientific literature. And this valuable data will be available for future researchers for comparison and further analysis, which is essential for conservation efforts.
(3) Manuscript preparation: The manuscript is not prepared properly. The title should provide a narrower area for reference as to where the Alakol Lakes are located than just saying Central Asia. Remember that international readers will not know where the lakes are located unless you atleast give the country name. Abstract is also not written properly and there is no flow to the arguments. Authors should provide background, aims of the study, how the study was conducted, what results were obtained and implications of the results in very brief to make a good abstract. Authors should also know that the threat categories are provided by International Union for Conservation of Nature (IUCN) RedList of threatened species and not just “International Red List”. Authors also mention change in “taxonomic” composition when they mean to say species composition. Remember, your work is on species diversity and composition. Taxonomic composition is a completely different topic in systematic biology, which has nothing to do with species composition in a given location. In Table 4, remember you are not providing “loads” of fish species but rather principle component factor loadings for the species. Figure 2 to 5 can be clubbed as a single figure with multiple panels for easy comparison for maps. References are not formatted properly.
Author Response
Dear Reviewer, many thanks for your time, patience and advice. We attached our point-by-point response below.

Reviewer 2 Report
This manuscript provides important updates on the taxonomic diversity of fish from the Alakol Lakes as well as an investigation of the assemblage information and abiotic factors that may predict their presence throughout the region. In this way the manuscript makes an important contribution to these important lakes. It is also helpful to connect the current collection information with historical data.
While the analyses are thorough, the interpretation of each of them is relatively sparse. The manuscript would be much more useful is there was more interpretation of the relationships (or lack of relationships) observed (particularly in Figures 6 and 7). For readers, many of which will be unfamiliar with the ecology of the Alakol Lakes would benefit from more detailed interpretations or inferences about why particular native species are more likely to be found in certain areas, or which non-native species are able to thrive under which conditions. This would provide potentially more actionable conclusions from this study.
One concern with this work is that the sampling effort took place in relatively small areas and included only kick net and landing nets. How representative of the broader study area are each of these sampling units? One hundred square meters is only a 10mx10m area, which seems rather small to be representative of near shore communities in the lake sites, and although the manuscript states that representative habitats of rivers were sampled, 100m2 seems a small area to effectively capture this at each river site as well. While the prohibition on electrofishing is understandable, why weren’t additional traditional fisheries sampling techniques used, including gill nets or traps used to sample in conjunction with active sampling techniques?
On Page 3, last sentence of 2.1. states,” A total of 50 fish samples were taken from 20 localities.” This could easily be taken to mean only 50 individual fish were taken from 20 localities, but based on Table 3, it’s clear that many more fish were sampled. Perhaps this should read, “a total of 50 sampling events occurred from 20 localities.”
Table 4: “Loads” should be “loadings” in the caption.
Page 9: is there any additional interpretation of the finding that Severtsov’s loach and the listed non-native fishes available were the principal drivers of differences? Would this analysis look different if the non-native fishes were excluded? Was this attempted?
Figure 7: It is difficult to interpret all three principal component axes in a single figure, particularly given the clustering and lack of site names for most points. This information would be more easily interpreted in three biplots (PC1 vs PC2, PC1 vs PC3, and PC2 vs PC3), rather than trying to disentangle them in one. This would perhaps allow for more clear understanding of the spatial patterns represented by the fish.
Page 2, below table 1 and Page 13 second paragraph, “spotted sloth beetle” is perhaps Spotted thicklip loach? This should be clarified.
For Figures 2-5 (which don’t appear to be numbered in the order in which they are referenced in the text), how were the spatial mapping of the nutrient concentrations performed? How was the polygon determined and how was the spatial gradient interpolated between what seems to have been point measurements?
Page 12: The paragraph which reads, “The Shagyntogay and Zhamanty rivers was going to be dry,” needs further interpretation and discussion, or it should be incorporated into another paragraph.
The last paragraph presents new information and interpretation of the results that deserve introduction earlier in the manuscript and more complete discussion. These are potentially interesting ideas that would benefit from further treatment in the text.
Author Response
Dear Reviewer, we are grateful for your time, patience and advice.Our point-by-point response is attached below.
